# Vaccination against PD-L1 with IO103 a Novel Immune Modulatory Vaccine in Basal Cell Carcinoma: A Phase IIa Study

**DOI:** 10.3390/cancers13040911

**Published:** 2021-02-22

**Authors:** Nicolai Grønne Jørgensen, Jeanette Kaae, Jacob Handlos Grauslund, Özcan Met, Signe Ledou Nielsen, Ayako Wakatsuki Pedersen, Inge Marie Svane, Eva Ehrnrooth, Mads Hald Andersen, Claus Zachariae, Lone Skov

**Affiliations:** 1National Center for Cancer Immune Therapy (CCIT-DK), Department of Oncology, Herlev and Gentofte Hospital, University of Copenhagen, 2730 Herlev, Denmark; jacob.handlos.grauslund.01@regionh.dk (J.H.G.); Ozcan.Met@regionh.dk (Ö.M.); Inge.Marie.Svane@regionh.dk (I.M.S.); Mads.Hald.Andersen@regionh.dk (M.H.A.); 2Department of Hematology, Herlev and Gentofte Hospital, University of Copenhagen, 2730 Herlev, Denmark; 3Department of Dermatology and Allergy, Herlev and Gentofte Hospital, University of Copenhagen, 2900 Hellerup, Denmark; Jeanette.Kaae@regionh.dk (J.K.); Claus.Zachariae@regionh.dk (C.Z.); Lone.Skov.02@regionh.dk (L.S.); 4Department of Immunology and Microbiology, University of Copenhagen, 2200 Copenhagen, Denmark; 5Department of Pathology, Herlev and Gentofte Hospital, 2730 Herlev, Denmark; Signe.Ledou.Nielsen@regionh.dk; 6IO Biotech ApS, 2200 Copenhagen, Denmark; awp@iobiotech.com (A.W.P.); ee@iobiotech.com (E.E.)

**Keywords:** vaccine, basal cell carcinoma, PD-L1, immunotherapy, clinical trial

## Abstract

**Simple Summary:**

Basal cell carcinoma is the most common skin cancer and new treatments for patients with widespread and numerous tumors are lacking. In a previous study treating patients with multiple myeloma with a peptide vaccine called IO103 against an immune checkpoint molecule called programmed death ligand 1, two cases of basal cell carcinoma regressed. The aim of the present study was to assess the effect of vaccination with IO103 in ten patients with basal cell carcinoma. Patients were vaccinated with Montanide as adjuvant up to nine times during six months. Regression in tumor size of at least 30% was seen in five of 18 tumors, two of which showed complete regression. Vaccinations resulted in immune responses against the vaccine in blood samples from nine of ten patients and in skin samples from five of nine patients. The findings suggest that the vaccine may be effective against some basal cell carcinomas.

**Abstract:**

Antitumor activity of immune checkpoint blocking antibodies against programmed death 1 (PD-1) in basal cell carcinoma (BCC) has been described. IO103 is a peptide vaccine against the major PD-1 ligand PD-L1. A phase IIa study of vaccination with IO103 and Montanide adjuvant was conducted in patients with resectable BCC (NCT03714529). Vaccinations were given six times every 2 weeks (q2w), followed by three vaccines q4w in responders. Primary endpoints were clinical responses of target tumors, change in target tumor size and immune responses to the vaccine. Secondary endpoint was safety. One tumor per patient was designated target tumor and biopsied twice during the course of vaccination. Synchronous non-target BCCs were not biopsied during vaccinations. Ten patients were vaccinated (six patients received six vaccinations and four patients received nine vaccinations). A partial response (PR) was seen in two target tumors. Two complete responses (CR) and one PR were observed in eight non-target tumors in four patients. No tumors progressed. Related adverse events were grade 1 and reversible. Immune responses against IO103 were induced in blood samples from nine of ten and skin-infiltrating lymphocytes from five of the nine patients. The regressions seen in non-target tumors suggest that IO103 may be effective against a subtype of BCC.

## 1. Introduction

Basal cell carcinoma (BCC) is the most common skin cancer [1]. Risk factors for BCC include sun exposure, advanced age and immunosuppression [2]. Most patients with BCC are successfully treated with locally directed therapies. A group of patients have frequent BCC that require cumulatively disfiguring surgery [3]. For the small group of patients with advanced BCC and patients with the rarely occurring metastatic BCC have very few approved treatment options beyond systemic hedgehog inhibitors. Several case reports of patients with advanced or metastatic BCC who have been successfully treated with immune checkpoint blocking antibodies targeting programmed death-1 (PD-1) have been published [4,5,6,7,8,9]. These reports found PD-L1 expression on both tumor cells and tumor-infiltrating lymphocytes in BCC. Thus, immunotherapy targeting the PD-1/PD-L1 axis can be effective, but the autoimmune side effects of immune checkpoint blocking antibodies hampers their use for the indolent majority of BCCs

Healthy donors and cancer patients harbor T cells which are specifically reactive to immune checkpoint molecules, including PD-L1 [10] (Munir et al. 2013a). Whereas monoclonal antibodies against PD-1 or PD-L1 work by binding to the cell surface portion of these molecules, the PD-L1 specific T cells are reactive against cells that are presenting PD-L1 derived peptide epitopes in the context of major histocompatibility (MHC) molecules (PD-L1-peptide: MHC complexes). Previous studies described a particularly immunogenic 19-amino acid peptide from PD-L1 which was named IO103. Stimulation with IO103 activates PD-L1 specific T cells to become pro-inflammatory, and cytotoxic to PD-L1 positive tumor cells and immune cells [11,12,13,14].

In a phase I study where IO103 vaccination with Montanide ISA-51 adjuvant was evaluated in patients with multiple myeloma, two patients with concomitant BCC experienced regression and clearance, respectively, of facial BCCs during vaccination with IO103 (NCT03042793) [15]. The patient whose BCC cleared completely, was in a state of complete remission (CR) from multiple myeloma at initiation of vaccinations but experienced a biochemical relapse of multiple myeloma during vaccinations. Interestingly, the biochemical relapse of myeloma coincided with reappearance of the BCC. This led us to hypothesize that immunologic control was shared between both the BCC lesion and the multiple myeloma. To explore whether vaccination with IO103 could have an effect in BCC, the phase IIa study presented here was conducted.

## 2. Results

### 2.1. Patient Characteristics

Ten patients, four females, six males, were recruited between 19 November 2018 and 21 October 2019 from the Department of Dermatology and Allergy, Herlev and Gentofte Hospital, Copenhagen, Denmark. Two patients had a history of multiple BCC and two patients were in treatment with low dose methotrexate due to chronic hand eczema or psoriasis (Table 1). All patients had at least one BCC, which was chosen to be target tumor. Four patients (patient #4, #5, #7 and #9) had more BCCs than the target tumor, the remaining tumors were called non-target tumors.

### 2.2. Clinical Outcome

#### 2.2.1. Target Tumors

Among the ten target tumors in ten patients, two tumors (20%) decreased at least 30% in longest diameter (PR), and eight tumors (80%) showed stable disease (SD) (Table 2). A decline in longest diameter was seen in 70% (seven of ten tumors) (Figure 1).

#### 2.2.2. Non-Target Tumors

Among the eight non-target tumors in four patients two CRs and one PR was seen. SD was noted for the remaining five tumors (Table 3). In non-target tumors 75% (six of eight tumors) decreased in size (Figure 2a–c and Figure 3).

#### 2.2.3. Overall Tumor Response

The overall level of response per patient is summarized as change in the sum of largest diameters. Overall, two patients (20%) reached PR and eight patients (80%) remained in SD (Appendix A). In the patients with more than one tumor, great variability of the level of change in tumor size was seen within the same patient (Figure 1b).

Patient #7 had four sBCCs that were all histologically verified as BCCs prior to initiation of vaccinations. On the day of the fourth vaccination, the target lesion had decreased 9%. The patient postponed the fifth vaccination for 81 days due to vacation, during which the target tumor had increased to the baseline size. Concurrent with the next two vaccinations, the size was again slightly (2 mm) reduced (Figure 2b).

Patient #9 had three sBCCs that were histologically verified prior to initiation of vaccinations. One BCC (largest diameter 12 mm) regressed completely between the day of biopsies and the initiation of vaccinations. The two other BCCs remained at enrollment and initiation vaccinations. The target tumor remained unchanged in largest diameter despite being biopsied after the second vaccination. The remaining non-target BCC regressed completely after the fifth vaccination, 145 days after being biopsied (Figure 2c). 

A special situation occurred regarding patient #4. The patient had a history of severely sun-damaged skin, with multiple actinic keratoses and frequent locally directed therapies against BCC. The patient was enrolled in this study with a treatment-naïve superficial BCC on the lower leg. After enrollment, relapse of a histologically verified BCC on the forehead which had been treated with curettage seven months earlier was noted, but regrettably not measured (Figure 3b and Appendix A). The histology had at the time of curettage shown nodular BCC with moderate to severe dysplasia but no invasive squamous cell carcinoma. Since the patient was already enrolled into this trial, the planned surgery was postponed until after the vaccination course was completed. The patient consented to a watch and wait approach. The treatment-naïve target tumor on the lower leg was biopsied after second and sixth vaccinations and was SD (regression 3 mm) during vaccinations (Figure 3a). After the eighth vaccination, the relapsed element on the forehead had regressed completely and planned surgical removal of this tumor was cancelled (Figure 3c). At 3 months follow-up, the BCC had not relapsed.

During the course of vaccinations, one patient observed improvement of folliculitis decalvans, one patient observed improvement of psoriasis and one patient experienced an objective and subjective improvement of porokeratosis. None of the lesions grew during vaccination. 

### 2.3. Adverse Events

Adverse events recorded during the course of vaccinations were of low grade. All patients experienced grade 1 reactions at injection sites (Appendix A). One patient (#4) had a history of decreased vision due to wet age-related macular degeneration (AMD) prior to inclusion. The condition had been treated with anti-vascular endothelial growth factor injections and photodynamic treatments, but due to lack of improvement, the patient had not received treatments against AMD for 1.5 years. During the course of vaccinations, the patient reported decreased vision on the right eye. The patients treating ophthalmologist examined the eye and confirmed that there was an increased involvement of the retina caused by AMD. The AE was not deemed related to vaccinations with IO103. 

### 2.4. Vaccine-Specific Immune Responses in Blood

IO103 vaccine-specific immune response in peripheral blood mononuclear cells (PBMC) was assessed by quantifying IFNγ-secreting lymphocytes by ELISpot at baseline, after two, six and eight vaccinations. An immune response (DFRx2) to IO103 in PBMCs was present in one patient at baseline. Responses (DFRx2) appeared after baseline in 70% of patients (7 of 10) (Figure 4a, raw data available in Appendix A). A modest but significant immune response (DFRx1) appeared in patient #5 and #6. Patient 8 had a response at baseline but not at the two later timepoints. Although all but one patient had DFR-defined responses post baseline, the mean increase in the amplitude of responses only reached statistical significance at the last timepoint with only four patients still receiving vaccines (Figure 4b).

### 2.5. Immune Responses in Skin-Infiltrating Lymphocytes

To assess immune reactivity against IO103, delayed type hypersensitivity (DTH) reaction with intradermal injections of IO103 without the adjuvant Montanide was performed [16]. DTH-injections were performed after the sixth vaccination. One patient did not wish to have DTH-injections. 48 h after DTH-injections, eight out of nine patients had an induration at the sites injected with IO103, which was at least double the size of the control injection.

Skin-infiltrating lymphocytes (SKILs) [17] were grown from biopsies of the IO103-injected DTH sites, and SKILs from five patients showed strong reactivity in IFNγ ELISPOT against IO103 (Figure 4c). High background precluded the evaluation of reactivity in two patients. SKILs from patient #6 did not grow to sufficient numbers.

### 2.6. Immune Phenotype in Peripheral Blood

Upon flowcytometric analysis of PBMCs, no changes met statistical significance. A tendency of declining proportion FoxP3^+^ Tregs and the subpopulation of CD15s^+^ Effector Tregs were noted during the vaccination course (Appendix A). No significant changes in T cell differentiation of CD8^+^ T cells and CD4^+^ T cells were seen (Appendix A).

### 2.7. Immunohistochemistry on BCC Biopsies

Upon immunohistochemical staining of biopsies from the target BCCs tumor cells did not express PD-L1 at baseline, while immune cells in the tumor frequently did (Appendix A, Representative examples of immunohistochemistry shown in Appendix A). During vaccinations one sample from the target tumor of patient #2 and the follow-up sample from patient #5 contained tumor cells which were PD-L1 positive (5–10% and 30%, respectively), all other samples contained PD-L1 negative tumor cells negative (data not shown). Interestingly, the original removed tumor from the forehead of patient #4 had 15% tumor cells positive for PD-L1. Tumor-infiltrating immune cells were variably PD-L1-positive during vaccinations (data not shown). Although more patients with changes in either target tumor or non-target tumor sizes during vaccinations had PD-L1 positive immune cells in target tumor biopsies, a clear relation was not found (Appendix A).

## 3. Discussion

In this study two PRs were seen in target tumors, and 70% of target tumors decreased a total of 12.8% in size during vaccinations. Only CTCAE grade 1 adverse reactions related to the vaccine such as injection site reactions were seen. All patients had histologically verified BCC. The target tumors were biopsied during the vaccination course, non-target tumors were not. In non-target tumors two complete responses and one partial response were reported. No tumors progressed in the treatment period. When a sum of tumor sizes was performed, 20% of patients (2 of 10) attained a PR.

One of the two CR was of a relapse of BCC, from which biopsies had not been taken. Interestingly, an increase in PD-L1 expression has been reported in relapsed BCC as compared to treatment-naïve tumors [5]. This could implicate that relapsed BCCs harbor a more inflammatory microenvironment in which stimulated PD-L1 specific T cells would encounter targets more frequently. Albeit uncommon, spontaneous regressions in BCC are a known phenomenon, with several published cases of partial regressions [18,19,20]. Spontaneous complete regressions of BCC are, however, inherently a difficult subject for scientific study [21].

The significance of immune surveillance to the control of BCC can be inferred from the greatly increased incidence in patients who are medically immunosuppressed due to organ transplantation compared with the incidence in the rest of the population [22]. Furthermore, BCC is like malignant melanoma one of the most highly mutated tumors and a high mutational burden has been correlated with response to immune therapy [23,24]. Immune reactions against IO103 was only found in 20% of patients and of low amplitude at baseline. In a study of patients with malignant melanoma, 33% of patients had immune reactions to IO103 in peripheral blood at baseline [25]. A bigger sample size would be needed to establish whether patients with BCC have a lower spontaneous response to IO103. Nonetheless, immune responses were present in blood samples in only two patients at baseline but found in in all but one patient during or after vaccinations.

Interestingly, a declining size of three non-target tumors in two patients (patient #5 and #7, Figure 2a,b) began after the respective patient’s target tumors had been biopsied, suggesting an “abscopal-like” effect on these non-target lesions. Regression of BCC after incomplete surgical removement the same element has been described [26]. While taking a biopsy from a tumor will likely induce an inflammation that could initiate a response to that same tumor, we have no data on whether biopsies of one BCC can affect others. Biopsies could activate damage-associated molecular patterns (DAMPs). This in turn would lead to a Th1-immune activation, and biopsies might in this manner have acted as an adjuvant to the vaccine. In future studies, a less invasive treatment with the approved toll-like receptor (TLR) activator imiquimod could be used instead as an additional adjuvant to vaccination of BCC.

In this study, the limited number of patients did not permit correlation of BCC subtypes to magnitude or type of response to treatment. This would be an interesting correlation which should be performed in future studies.

Both patients with complete responses also had tumors with no or very discrete decrease in size (Figure 1b). A heterogenicity of tumors in terms of genetics or immune microenvironment could account for these differences. With the small patient cohort, no formal correlation to decreases in tumor sizes could be made to immune infiltration in target tumors (Appendix A) or tumor immune cell PD-L1 expression (Appendix A). Nonetheless, it is interesting that responses were seen although almost all biopsies showed that BCC tumor cells themselves were PD-L1 negative, and only the immune cell compartment was PD-L1 positive. This illustrates the immune modulatory nature of the vaccine and how this vaccine approach distinguishes from traditional cancer vaccines, that target tumor specific antigens. A similar observation was recently obtained with another immune modulatory vaccination approach against the immunosuppressive enzyme indoleamine 2,3-dioxygenase 1 (IDO1). This study illustrated that IDO1 vaccination effectively induced anti-tumor immune responses in an in vivo model of cancer where the IDO1 expression was exclusively limited to tumor-infiltrating immune cells and not tumor cells [27]. It would be interesting to examine PD-L1 on immune cells circulating in peripheral blood since the direct impact on PD-L1 expressing cells that follows the activation of PD-L1 specific T cells is counteracted by inflammation-induced PD-L1 expression on other cells. This will be part of future studies on vaccination with IO103.

## 4. Subjects and Methods

### 4.1. Study Design

This study was a one-armed phase IIa study conducted in collaboration between Department of Dermatology and Department of Oncology, Herlev and Gentofte Hospital. The study was conducted in accordance with the Helsinki Declaration, and Good Clinical Practice (GCP) recommendations. All participants gave written informed consent before enrollment. The protocol was approved by the Ethics Committee of the Capital Region of Denmark 23 October 2018 (H-18032478), the National Board of Health 12 September 2018 (EudraCT 2018-002605), and the Danish Data Protection Agency 16 August 2018 (VD-2018-265). Key eligibility criteria included biopsy verified BCC ≥ 14 mm in longest diameter at the time of screening, no previous treatment with SHH, no severe autoimmune diseases or active infections. Full in- and exclusion criteria can be found in Appendix A.

### 4.2. Treatment

Subcutaneous vaccinations contained 100 µg IO103, a 19-amino-acid peptide (FMTYWHLLNAFTVTVPKDL) from the signal peptide of PD-L1 (PolyPeptide Laboratories, Strasbourg, France). The peptide was dissolved in dimethylsulfoxide (DMSO), sterile filtered, and frozen at −20 °C (NUNC^™^ CryoTubes^™^ CryoLine System^™^ Internal Thread, Sigma-Aldrich, St. Louis, MO, USA). A maximum of two hours before administration, the peptide was thawed and dissolved in sterile water for injection. Immediately before injection, the dissolved peptide was emulsified 1:1 with the adjuvant Montanide ISA-51 (Seppic Inc., Paris, France) to form a total volume of 1 mL [28]. Vaccinations were administered every two weeks for 6 times with the possibility of extension with three additional vaccines given monthly. Thus, up to a total of 9 vaccinations could be given. Residual tumors were surgically removed.

### 4.3. Clinical Evaluation

All tumors were biopsied before initiation of vaccination (21–95 days prior to first vaccination). Target tumors were also biopsied after two and six vaccinations, but non-target tumors were not biopsied during the course of vaccinations.

The study had three primary end points:Clinical response of the target BCC. Responses were defined as clearance: 100% reduction in tumor; partial response: 30–99% reduction in tumor; worsening: more than 20% increase in tumor; no response: none of the above.Disease control rate. Defined as the magnitude of reduction of the largest diameter of the target BCC after 6 treatments with IO103.Immune responses in biopsies from BCC after treatment with IO103.

The study had two secondary endpoints:Immune responses in skin after delayed type hypersensitivity injections of IO103.Incidence of treatment emergent adverse events (safety and tolerability).

Since no standard for response assessment to therapy in patients with BCC has been established, the response definitions were adapted from previous studies of systemic therapy on BCC [29,30]. Changes in longest diameter are used to describe the responses. All measurements of tumors are shown in Appendix A. The description of responses in non-target lesions was not predefined in the protocol but are described as an exploratory outcome. 

At the time of diagnosis four tumors were measured by the principal investigator (PI), and tumors on the remaining six patients were measured by 6 doctors. For the measurements at diagnosis, some interindividual variation is to be expected, but importantly baseline measurements at the first vaccination, measurements during vaccinations and at evaluation after last vaccine, were all performed by one person, the PI. Adverse events were graded according to CTCAE version 4.01 at every visit. 

### 4.4. Tumor Biopsies

Tumors were histologically verified as BCC on 3 mm punch biopsies. Additional biopsies were taken from the target tumors after two and six vaccinations. Immunohistochemical automated staining was performed using the anti-CD3 clone F7.2.38, anti-CD8 clone C8/144B and anti-PD-L1 clone 28-8 (all Agilent Technologies, Santa Clara, CA, USA). Furthermore, immunohistochemical assessment of densities of T lymphocytes was performed by HalioDX, Marseille, France using primary antibodies (CD3, HDX1, CD8, HDX2 or PD-L1, HDX3) and detection with a secondary antibody (ultaView Universal DAB Detection Kit, Roche, Basel, Switzerland, catalog #760-500).

### 4.5. Blood Samples

Blood samples for isolation of serum and peripheral blood mononuclear cells (PBMCs) were obtained at baseline, after two vaccinations (on the day of the third vaccination), at the day of the sixth vaccination and at the day of the ninth vaccination. PBMCs were isolated by gradient centrifugation of heparinized blood on Lymphoprep (STEMCELL Technologies, Vancouver, BC, Canada) in LeucoSep tubes (Greiner Bio-One, Kremsmünster, Autria). Isolated PBMCs were cryopreserved in 90% human serum (Sigma-Aldrich) with 10% DMSO (Sigma-Aldrich).

### 4.6. Delayed Type Hypersensitivity and Skin Infiltrating Lymphocytes

The presence of vaccine-reactive cells at sites of DTH-injections was assessed after six vaccinations. On the lower back, three intradermal injections of IO103 without adjuvant and one control injection of aqueous solvent containing DMSO without peptide or adjuvant was administered. After 48 h post-DTH injection, skin reaction was measured, and 4 mm diameter punch biopsies were taken from the sites of IO103-containing injections and cut into fragments. The fragments were cultured in 24-well plates for 3–5 weeks in RPMI-1640 with 10% human serum and 100 U/mL interleukin-2 (IL-2) with penicillin, streptomycin, and fungizone. Three times weekly, half the medium was replaced with fresh medium containing IL-2. Skin-infiltrating lymphocytes emigrated from the biopsies were harvested after 3–5 weeks in culture to be tested in ELISPOT assays (see Section 4.7 below). The remaining lymphocytes were cryopreserved, as described for PBMCs.

### 4.7. IFNγ ELISpot Assay

To assess T-cell responses against IO103, we performed indirect interferon gamma Enzyme-Linked ImmunoSPOT (IFNγ-ELISpot) assays, as previously described [10]. PBMCs for both the negative control wells and positive wells were stimulated once in vitro to increase the sensitivity of the assay [31]. Briefly, cryopreserved PBMCs were thawed and stimulated once with IO103 at RT in 24-well plates with 0.5 mL X-VIVO medium. Two hours after, 1.5 mL X-VIVO medium with 5% human serum was added, and the plate was incubated at 37 °C with 5% CO_2_. The next day, IL-2 was added, yielding a concentration of 120 U/mL. After 8–14 days, stimulated PBMCs were added with or without IO103 in a 96-well nitrocellulose plate (MultiScreen, MAIP N45; Millipore) which had been precoated with anti-IFNγ-mAb (mAb 1-DIK, Mabtech, Sweden) and the cells were incubated overnight. The day after, the plates were washed, biotinylated secondary anti-INFγ mAb (Mabtech) was added, and the plates were incubated for two hours at RT. The plates were washed, Streptavidin-enzyme conjugate (AP-Avidin; Calbiochem/Invitrogen Life Technologies, Waltham, MA, USA) was added, and the plates were incubated for one hour at RT, and then washed. The enzyme substrate NBT/BCIP (Invitrogen Life Technologies) was added. Resulting spots were counted using the ImmunoSpot Series 2.0 Analyser (CTL Analyser, Cleveland, OH, USA). Maximum count was set to 500 spots/well. IFNγ-ELISPOT assays on PBMCs were run in triplicate with 2.5–3.5 × 10^5^ cells/well. IFNγ-ELISPOT assays on SKILs were run in triplicate or quadruplicate with 3 × 10^5^ cells/well, using a reversed sequence of the IO103 peptide as a control. No positive controls were used.

### 4.8. Flow Cytometry on PBMCs

Flow cytometry on PBMCs was performed as previously described [15], for details see Text S1. 

### 4.9. Statistical Analysis

The nature of the phase IIa clinical trial was exploratory. Since no previous trials have studied a similar vaccine in patients with basal cell carcinoma, no formal power calculations could be performed. Based on previous experience with similar vaccines, a sample size of ten patients was deemed sufficient to evaluate safety and explore effects on paraclinical tumor markers. All participants enrolled and were treated in accordance with the protocol were included in the statistical analysis.

Responses in ELISpot were assessed in tests with more than six spots per 1 × 10^5^ cells per well with the distribution-free resampling (DFR(2x)) method as described by Moodie et al. [32]. Student’s *t* test was used to assess responses in samples run in duplicates. When comparing differences in ELISpot responses or flow cytometric subpopulations the repeated measures ANOVA test or mixed-effects model using the restricted maximum likelihood method if missing values, both with Tukey’s multiple comparisons test was used for comparing baseline to after second and after sixth vaccination. Due to only four patients proceeded to receive nine vaccinations, a pairwise T test was used to compare the four datapoints after nine vaccinations to their corresponding baseline controls. A *p*-value of ≤0.05 was considered significant.

## 5. Conclusions

In conclusion, vaccination with IO103 in BCC was safe with very limited adverse events. Vaccination led to induction of immune responses in eight of nine patients. During the vaccination course, no progressions were seen, and tumor responses were seen in tumors that were biopsied as well as in tumors not biopsied during the treatment. The low toxicity and high immunogenicity of the vaccine shown in this study have supported the initiation of further studies. In skin cancer a larger trial is ongoing vaccinating patients with melanoma in combination with anti-PD1 antibodies.

## Figures and Tables

**Figure 1 cancers-13-00911-f001:**
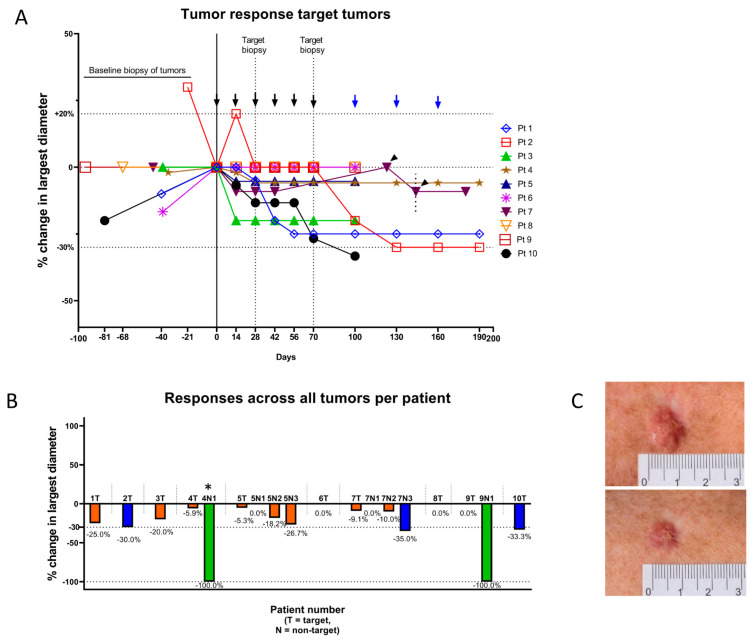
Tumor responses in target tumors. (**A**) Spiderplot of percent change in largest diameter. Tumor sizes normalized at first vaccination (day 0). Patient #1, #6 and #10 had increasing sizes from diagnostic biopsy until initiation of vaccinations. Arrows indicate vaccination with IO103. Six vaccines were given to all patients, while an additional three vaccines (blue arrows) could be given in patients depending on response. Thus, at day 100, patients 1, 2 and 4 received seventh vaccination and proceeded to receive in total nine vaccines while the remaining patients were evaluated after their sixth vaccination. Patient #7 had a treatment break after the fourth vaccination during which the tumor increased until restart of vaccinations (arrowheads indicate patient #7′s fifth and sixth vaccination). (**B**) Percent change from baseline in largest diameter for all tumors. *: The non-target tumor on patient #4 was not measured but only photographically documented before clearance after eighth vaccination. (**C**) Example of PR: target tumor of patient #10. Top: At baseline. Bottom: At evaluation after six vaccinations.

**Figure 2 cancers-13-00911-f002:**
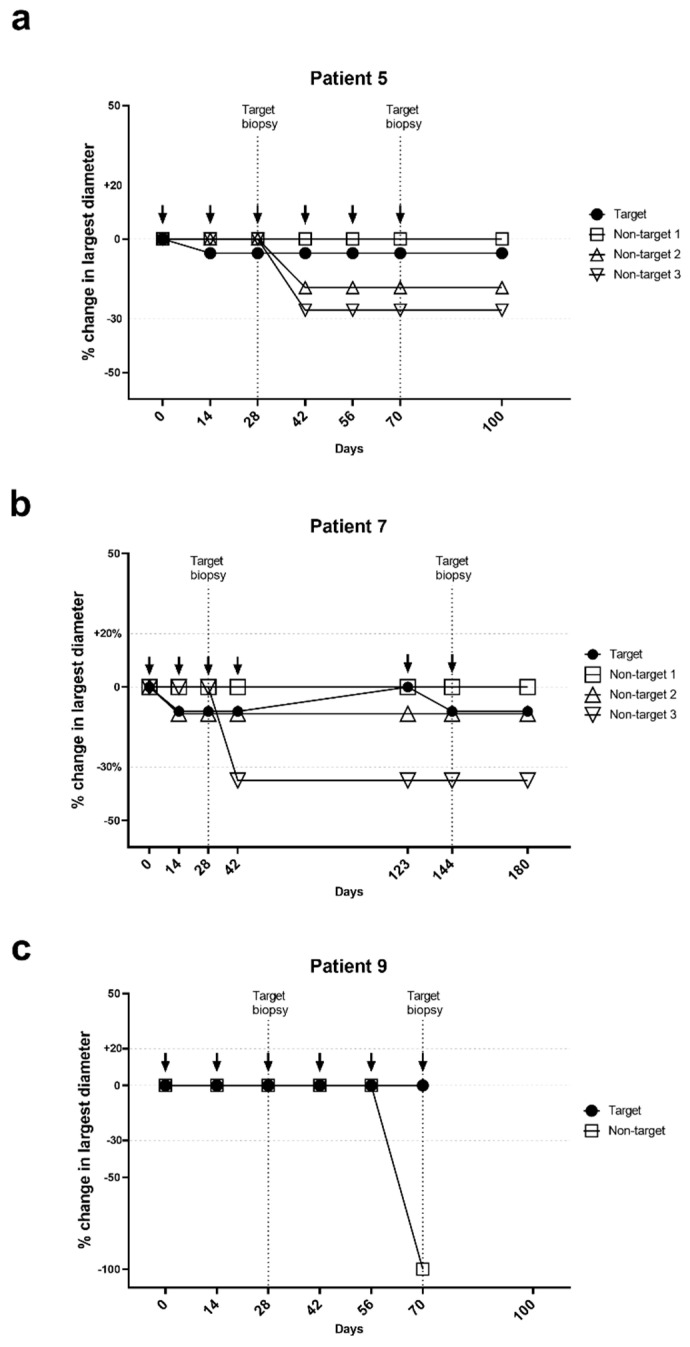
Tumor responses during vaccination with IO103 in patients #5, #7 and #9. Arrows indicate vaccination with IO103. (**a**) Patient #5, (**b**) Patient #7, (**c**) Patient #9.

**Figure 3 cancers-13-00911-f003:**
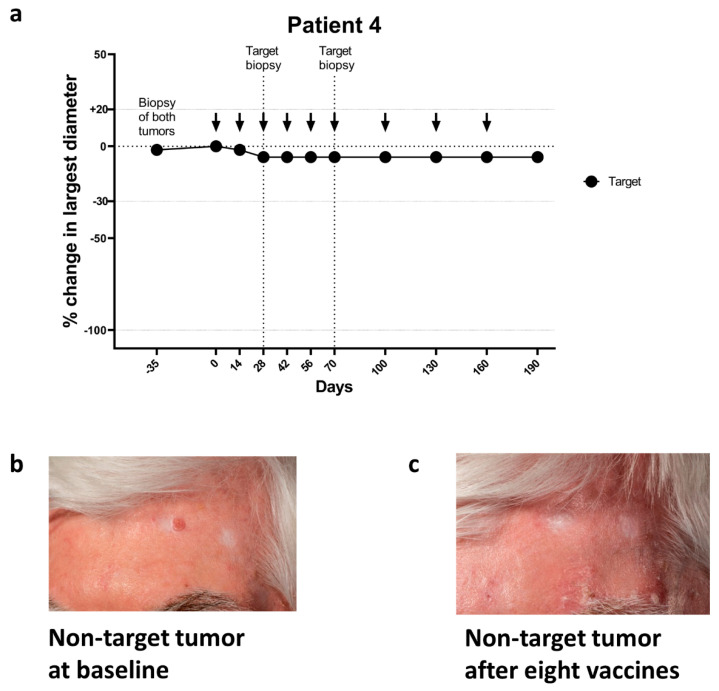
Tumor responses during vaccination with IO103 in patient #4. (**a**) Target tumor of patient #4. (**b**) Clinical photos of relapsed BCC on patient number #4, at baseline and (**c**) after eight vaccinations. This tumor was only documented by clinical examination and photo and was not measured before the visit at the ninth vaccine.

**Figure 4 cancers-13-00911-f004:**
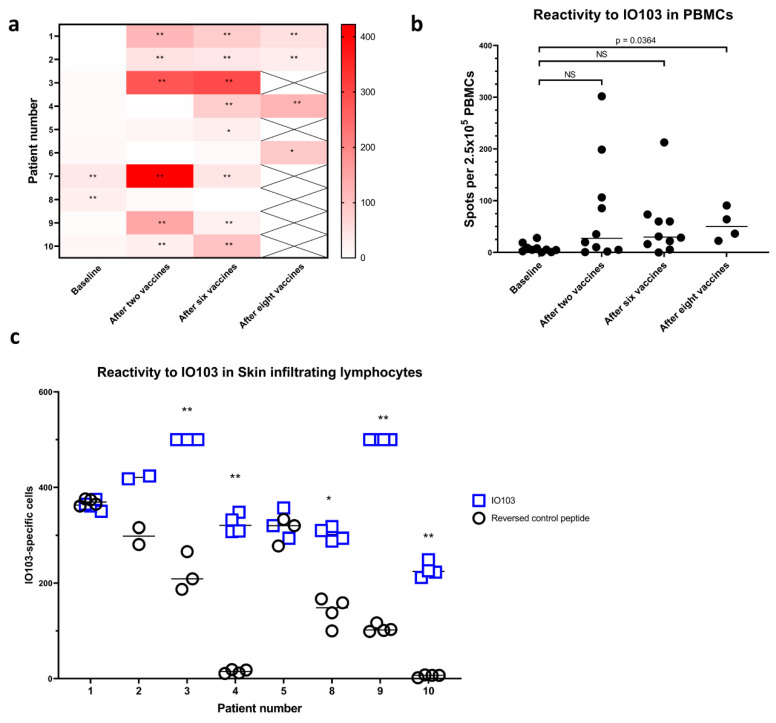
Immune responses. IFN-γ ELISpot reactions against IO103, background subtracted. (**a**) Immune responses in PBMCs per patient. Spots per 2.5 × 10^5^ in patients #1 to #3 and spots per 3.5 × 10^5^ in patients #4 to #10. *: DFRx1, **: DFRx2 (**b**) Immune responses in PBMCs per timepoint, samples from patient #4 to #10 have been normalized to spots per 2.5 × 10^5^. For comparisons baseline to after two and after six vaccines, repeated measures ANOVA with Tukey’s multiple comparisons test was used, for comparison of baseline to after eight vaccines, paired *t* test was used, for both tests *p* ≤ 0.05 considered significant. (**c**) Immune responses in skin-infiltrating lymphocytes. SKILs from patient #6 did not grow sufficiently. Patient #7 did not have DTH performed. SKILs tested in IFN-γ ELISpot against IO103 or a scrambled control peptide. SKILs per well: Pt #1: 3.0 × 10^5^, Pt #2: 1.0 × 10^5^, Pt #3: 1.2 × 10^5^, Pt #4: 2.85 × 10^5^, Pt #5: 1.7 × 10^5^, Pt #8: 2.0 × 10^5^, Pt #9: 2.25 × 10^5^, Pt #10: 2.6 × 10^5^. *: DFRx1, **: DFRx2. SKILs from patient #2 were tested in duplicate, a Student’s *t* test *p* = 0.11. NS = non-significant.

**Table 1 cancers-13-00911-t001:** Patient characteristics.

ID	Sex	Age	Previous Skin Cancer	Co-Morbidity	Immuno-Suppressive Medications
1	Male	50	BCC × 33. Superficial MM × 1	BRCA-1 and BRCA-2 mutatet, Apoplexia cerebri, Retinal detachment, Malignant melanoma, Thyroiditis, Diabetes type II, Folliculitis decalvans, Hypercholesterolemia, Hypertension	None
2	Female	47	None	None	None
3	Female	56	None	Hypertension	None
4	Male	73	BCC × 5	Age-related macular degeneration (AMD), Allergic hand eczema	Methotrexate 15 mg weekly
5	Female	69	None	Psoriasis	None
6	Male	76	None	Porokeratosis	None
7	Female	56	None	Slipped disk	None
8	Male	59	None	Hypertension, Psoriasis, C. prostatea operata	Methotrexate 15 mg weekly
9	Male	73	BCC × 29	Hypertension	None
10	Male	72	BCC × 2	Hypertension	None

BCC: Basal cell carcinoma. MM: malignant melanoma.

**Table 2 cancers-13-00911-t002:** Target tumor characteristics and responses Tumor characteristics and final response after vaccination course. The two non-target BCCs on patient one were not followed during the study. Patient nine had three biopsy verified BCC at diagnosis, but one could not be found at start of vaccinations. High Background indicates that the samples had high background which precluded evaluation of response to IO103.

ID	Target Tumor Location	Target Tumor Type (Longest Diameter, mm)	Target, Clinical Response	Target, Change (%)	ELISPOT Response to IO103 in PBMCs	ELISPOT Response to IO103 in SKILs
1	Shoulder	sBCC (20)	NC	−25	**	High Background
2	Chest	sBCC (10)	PR	−30	**	0
3	Upper arm	nBCC (25)	NC	−20	**	**
4	Lower leg	sBCC (51)	NC	−5.9	**	**
5	Back	sBCC (19)	NC	−5.3	*	High Background
6	Chest	nBCC (18)	NC	0	*	NA
7	Back	sBCC (22)	NC	−9.1	**	NA
8	Shoulder	nBCC (14)	NC	0	0	*
9	Back	sBCC (20)	NC	0	**	**
10	Shoulder	nBCC (15)	PR	−33	**	**

CR: complete response, nBCC: nodular BCC, NA: not available, NC: no change, PR: partial response, sBCC: superficial BCC, *: DFRx1, **: DFRx2.

**Table 3 cancers-13-00911-t003:** Non-target tumor responses Final response after vaccination course.

ID	Non-Target BCCs n (Longest Diameter, mm)	Non-Target, Clinical Response	Non-Target, Change (%)
4	1 (not measured)	CR	−100
5	3 (15, 13, 11)	NC	0
		NC	−18.2
		NC	−26.7
7	3 (20, 12, 10)	NC	0
		NC	−10
		PR	−35
9	1 (6)	CR	−100

CR: complete response, nBCC: nodular BCC, NC: no change, PR: partial response, sBCC: superficial BCC.

## Data Availability

Further data available upon request from the corresponding author. The data are not publicly available due to privacy of the subjects.

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
