# Peer review of "Vaccination against PD-L1 with IO103 a Novel Immune Modulatory Vaccine in Basal Cell Carcinoma: A Phase IIa Study"

_cancers, 2021, doi:10.3390/cancers13040911_

Round 1
Reviewer 1 Report
Summary of the main findings of the study
The article by Jørgensen and colleagues presents the data of a phase IIa clinical trial of the efficacy and tolerability of the IO103 vaccine tested on 10 patients with basal cell carcinoma. IO103 is a peptide vaccine against PD-L1 administered subcutaneously to the patients as an emulsion in Montanide ISA-51. The authors first describe the clinical outcome and the evolution of the size of the target and non-target tumors during the course of the vaccination, as well as the adverse events. They then describe the specific systemic T cell response in the blood of patients with an ELIspot test carried out using PBMCs pre-stimulated with the peptide used in the vaccine. Thus, the local cellular immune response is analyzed using the DTH reaction as well as an ELISpot test carried out using infiltrating lymphocytes obtained from tumor biopsies. Finally, the phenotype of peripheral T cells as show by flow cytometry and the expression of PD-L1 within the tumors using immunohistochemistry staining are briefly described.
The authors report a clear antitumor effect and a significant immune response, however there are some critical points and errors of interpretation as well as the absence of essential data as to the formulation of the conclusions, to be able to accept this article in Cancers before a major revision.
Comments on the methods, results and data interpretation
Importants comments and observations
- There is a lot of ambiguity in the design of the vaccination, as some patients did not receive the 9 doses initially intended, and yet, although this is indicated in the legend to the figures, the data are presented in a misleading manner as if all donors could be compared together, and these deviations in the clinical trial are little discussed. This lead to erroneous figures as in Figure 1 where the indicated treatment scheme does not correspond to the actual treatment received by most of the patients. In my view, it is urgent to separate the patients into comparable groups in all the analyzes described in the article, from the anti-tumor response to the systemic and local immune response, despite the limited number of patients in the study. Lack of vaccine doses could lead to important cellular and clinical effects and should be considered a major factor between different treatment groups.
- Another critical point is the case of the non-target tumor in patient #4 analyzed in Figure 2 and Table 3. As seen in the Supplementary Table 2, this tumor was not measured until the 9th vaccination. Therefore the reader has no idea about the size of this tumor at the initiation of the study, and more problematic is the clinical outcome defined as “complete response” by the authors with -100% volume change. It is not acceptable to make such conclusions in the absence of data, therefore this case should be presented differently. Moreover the skin of patient #4 appears very different between before and after vaccination and the skin lesions, in particular around the ocular orbit, this should be discussed in the text in order to resolve any doubt.
- The analysis of the T response is of major importance in the study of the clinical and immunological effects resulting from peptide vaccination. This point is briefly discussed in the Results section but no data as to the activation phenotype of CD8+ and CD4+ T cells are shown. Authors should present these data at least in Supplementary Materials. This will also help to establish a relation with the results obtained in ELISpot. The same is true for the PD-L1 target. This is indeed reported in the Results section, in particular as regards the expression of PD-L1 on tumor cells, but no data as to the presence of circulating PD-L1 cells, in particular PD-L1-positive myeloid cells, which however seem to be the privileged target of an anti-PD-L1 T cell response induced by the vaccine. Thus the authors should analyze the expression of PD-L1 in circulating immune cells and discuss the data.
Minor points
- Line 78: “six females, four males” does not correspond to Table 1 which lists 6 males and 4 females. Please clarify.
- Figures 1-3: text annotations and axis labels are too small.
- Figure 2E,F: please date both images with respect to vaccinations.
- Figure 3: please provide the raw data for the ELISpot, i.e. without background subtraction.
- Figure 3B: repeated measures ANOVA with the appropriate post-hoc test is more appropriate in this analysis design as the authors compare the same groups at different time-points. I suggest the authors use repeated measures ANOVA for the full data sets at the 3 first time points, and a simple pairwise t-test to compare the 4 data points measured after 8 doses to their corresponding base line control.
- Line 203: please share some representative images from immunohistochemistry staining.
- Supplementary Figure 3: please confirm if the ANOVA test have been used.
- Line 234: “we have found immune reactions in healthy donors and patients with several types of cancer”, please precise the order of magnitude and if this is comparable to the data presented in the current article.
- Line 237: “immune responses were induced in blood samples in all but one patient”, what is the threshold used by the authors to make this conclusion? From my perspective, and based on Figure 3B, it looks rather like 4 patients out of 10 exhibited such an immune response. A statistical definition of this threshold, e.g. based on the baseline response, should be specified in the materials and methods.
- Lines 250-253: “No correlation between immune infiltration in target tumors and decrease in tumor size was seen (Supplementary Figure 4) and no correlation between immune cell PD-L1 expression and decrease in tumor size was seen (Supplementary table 4)”. The authors should performed proper correlation analyses.
- Lines 352-356: please confirm that the PBMCs used as background control have been also pre-stimulated with IO103 peptide and then stimulated with the reversed sequence peptide for IFNγ release. Please justify why the negative control is different in ELIspot with PBMCs and with SKILs? Also please confirm if any positive control was used.
Reviewer 2 Report
This is a very interesting and promising papers on phase II study on vaccination against PD-L1 using IO103 in BCC.
The major limit of the study is represented by the poor number of cases (and lesions) considered.
Several other minor comments:
- BCC subtypes and response to treatment correlation represents a really interesting point of the study and needs to be deeper considered as well as analysed
- in page 9, lines from 219 to 227: please better compare the available data on spontaneous regression of BCC and the immune response
- considering the small number of cases (as well as of total lesions) my advice is to report case by case description including response to the treatment (lesion target regression, subtypes of BCC and relative respinse to treatments, etc.) and the immunological background of the patient
- does the subtype of BCC correlates to the modality of response (reduction in diameter or flattening of the lesions?)
Reviewer 3 Report
In their manuscript, Jorgensen et al analyze efficacy and safety in a small cohort of patients with basal cell carcinoma (BCC) treated with a vaccine targeting a PD-L1-derived peptide. They report some clinical responses both in target and not target lesions, and induction of immune responses in an important percentage of vaccinated patients, while most adverse events were mild and reversible. This is an interesting strategy that deserves further consideration in larger groups. The following comments arise:
- Previous papers listed in the reference section describe HLA-restricted responses and associated epitopes in the 19-mer IO103 immunogen. Is there any criteria for patient selection related to expression of determined HLA molecules associated to these epitopes? Moreover, is there any association between clinical and immunological responses and HLA types?
- Have authors analyzed association between clinical responses and immune responses in blood and tumor?
- Authors indicate the lack of PD-L1 expression on tumor cells. Which would be the mechanisms of action in this setting? Have they considered the possibility of bystander T cells generated as a consequence of epitope spreading after vaccination?
- Authors hypothesize that PD-L1 expression may be due to a more inflammatory microenvironment. Do they see any association between PD-L1 expression and T cell infiltration?
- Finally, although the number of patients is small to draw general conclusions about overall efficacy, it seems that efficacy is limited. Would they propose any strategy to enhance the response rate and the magnitude of responses?
Minor comment:
Line 23, word “tumor” is duplicated
Round 2
Reviewer 1 Report
I thanks the authors for taking into consideration my recommendations and answering to almost all the questions I raised, which allowed to take away some doubts and to enhance the last but not least technical and statistical points. In particular, Jørgensen and colleagues were able to clarify the vaccination schedule, which appeared to be differential but which is now better justified. The authors also present the case of the non-target tumor in patient #4 as a separate case study, which will avoid confusion for the reader. Finally, I would like to thank the authors for providing more details on the systemic cellular immune response before and after vaccination and for discussing a little more the implication of PD-L1 which is ultimately the key target of the IO103 vaccine. I have no further point to add.